



# Mid- and Far-Infrared Spectral Signatures of Mineral Dust from Low- to High-Latitude Regions: significance and implications

Claudia Di Biagio[1], Elisa Bru[2], Avila Orta[2], Servanne Chevaillier[2], Clarissa Baldo[1,2], Antonin Bergé[1,a], Mathieu Cazaunau[2], Sandra Lafon[1], Sophie Nowak[3], Edouard Pangui[2], Meinrat O. Andreae[4,5,6], Pavla Dagsson-Waldhauserova[7,8], Kebonyethata Dintwe[9], Konrad Kandler[10], James S. King[11], Amelie Chaput[11,b], Gregory S. Okin[12], Stuart Piketh[13], Thuraya Saeed[14], David Seibert[15], Zongbo Shi[16], Earle Williams[17], Pasquale Sellitto[2,18], Paola Formenti[1]

[1]Université Paris Cité and Univ Paris Est Creteil, CNRS, LISA, F-75013 Paris, France
[2] Univ Paris Est Creteil and Université Paris Cité , CNRS, LISA, F-94010 Créteil, France

[3] Université Paris Cité, CNRS, ITODYS, Paris F-75013, France

[4] Max Planck Institute for Chemistry, 55128 Mainz, Germany

[5] Department of Geology and Geophysics, King Saud University, Riyadh, Saudi Arabia

[6] Scripps Institution of Oceanography, University of California San Diego, La Jolla, CA, USA

[7] Faculty of Environmental and Forest Sciences, Agricultural University of Iceland, 311 Hvanneyri, Iceland

[8] Faculty of Environmental Sciences, Department of Water Resources and Environmental Modeling, Czech University of Life Sciences Prague, 165 00 Prague, Czech Republic

[9] Department of Environmental Science, University of Botswana, Gaborone, Botswana

[10] Institute of Applied Geosciences, Technical University of Darmstadt, 64287 Darmstadt, Germany

[11] Université de Montréal, Québec, Canada

[12] Department of Geography, University of California – Los Angeles, Los Angeles, California, USA

[13] NorthWest University, Potchefstroom, South Africa

[14] Department of Physics, Lakehead University, Thunder Bay, Ontario, Canada

[15] Walden University, Minneapolis, Minnesota, USA

[16] School of Geography, Earth and Environmental Science, University of Birmingham, Birmingham, B15 2TT, U.K.

[17] Parsons Laboratory, Massachusetts Institute of Technology, Cambridge, Massachusetts, USA

[18] Istituto Nazionale di Geofisica e Vulcanologia, Osservatorio Etneo, Catania, Italy

[a] now at: Laboratoire des Sciences du Climat et de l'Environnement, CEA–CNRS–UVSQ, IPSL, Université Paris-Saclay, 91191 Gif-sur-Yvette, France

[b] now at: Department of Civil and Environmental Engineering, National University of Singapore, 117576, Singapore

*Correspondence to*: Claudia Di Biagio (Claudia.dibiagio@lisa.ipsl.fr)

**Abstract.** Mineral dust absorbs and scatters solar and infrared radiation, thereby affecting the radiance spectrum at the surface and top-of-atmosphere and the atmospheric heating rate. While half of the outgoing thermal radiation is emitted in the far infrared (FIR, 15-100 μm), knowledge of the optical properties and thermal radiative effects of dust is currently limited to the mid-infrared region (MIR, 3-15 μm). In this study we performed pellet spectroscopy measurements to evaluate





the MIR and FIR contribution to dust absorbance and explore the variability and spectral diversity of the dust signature within the 2.5-25 µm range. Thirteen dust samples re-suspended from parent soils with contrasting mineralogy were

investigated, including low and mid latitude dust (LMLD) sources in Africa, America, Asia, and Middle East, and high latitude dust (HLD) from Iceland. Results show that the absorbance of dust in the FIR up to 25 µm is comparable in intensity to that in the MIR. Also, spectrally different absorption (position and shape of the peaks) is observed for HLD compared to LMLD, due to differences in mineralogical composition. Corroborated with the few available literature data on absorption properties of natural dust and single minerals up to 100 µm wavelength, these data suggest the relevance of MIR and FIR

interactions to the dust radiative effect for low to high latitude sources. Furthermore, the dust spectral signatures in the MIR and FIR could potentially be used to characterise the mineralogy and differentiate the origin of airborne particles based on infrared remote sensing observations.

## 1 Introduction

Mineral dust is one of the most abundant aerosol species in the Earth's atmosphere with far reaching implications in the

climate system (Knippertz and Stuut, 2014; Kok et al., 2023). Over 4000 Teragrams of dust are emitted annually from low and mid-latitude large deserts in Africa, Asia, America, and the Middle East, here referred as LMLD (low and mid latitude dust) (Ginoux et al., 2012; Gliß et al., 2021; Kok et al., 2021, 2023). In addition, new high latitude dust (HLD) sources are progressively emerging in response to glacier and snow retreat due to global warming and contribute to dust load especially in the Arctic (Bullard et al., 2016; Meinander et al., 2022, 2025).

Mineral dust aerosols strongly affect the radiative budget at local, regional and global scale via scattering and absorption of both solar (0.2-3 µm) and infrared (3-100 µm) radiation (Li et al., 2004; di Sarra et al., 2011; Slingo et al., 2006; Song et al., 2022; Yang et al., 2009). The capacity of dust to interact with radiation throughout the atmospheric spectrum is due to its extended size distribution, including particles from hundreds of nanometres to several tenths of micrometres (Formenti and Di Biagio, 2024; Ryder et al., 2013), and its mineralogical composition, including silicates, carbonates and iron oxides

(Formenti et al., 2014b; Jeong, 2008) showing multiple absorption bands across the shortwave to infrared spectral range (Sokolik and Toon, 1999). In particular, strong infrared absorption is identified due to the resonance peaks of quartz, clays (illite, kaolinite, smectite), and calcite (Di Biagio et al., 2014b, 2017; Hudson et al., 2008b, a; Sokolik and Toon, 1999). The absorption in the infrared contributes to a positive dust direct radiative effect (DRE) at TOA, which opposes and counteracts a significant part of the dust negative DRE resulting from the dominant scattering at solar wavelengths (Christopher and

Jones, 2007; di Sarra et al., 2011; Song et al., 2022).

Recent modelling efforts including state-of-the-art representation of dust size distribution and spectral scattering and absorption from the ultraviolet to the mid-infrared MIR (3-15 µm) suggest that at the global and annual scale the dust-radiation interactions in the infrared can counteract up to 88% of the negative DRE at TOA (Di Biagio et al., 2020; Kok et



al., 2017; Wang et al., 2024). However, although the spectral range within 15 and 100 μm, referred to as far infrared (FIR),
is also of great relevance for the Earth's radiative budget, as about half of thermal radiation re-emitted by the Earth and
atmosphere towards space falls within this spectral range (e.g., Harries et al., 2008), the state of current knowledge is limited
to the MIR range, since the dust absorption signature and refractive index above 15 μm in the FIR remain unexplored.

A better knowledge of dust-radiation interactions in the FIR is key to fully understand, model and predict the role of dust
aerosols on the Earth's radiative budget, as well as for exploiting current and future satellite missions measuring in the MIR
and FIR. Indeed, the interaction of dust with infrared radiation modifies the radiance flux spectrum at the surface and the
TOA, therefore making particles detectable from space-borne and ground-based remote sensing sensors working at different
spectral ranges. Provided with accurate input data on particle spectral complex refractive index, remote sensing instruments
can use atmospheric observations in presence of suspended dust to get information on dust layers' physico-chemical
properties, such as their size distribution, optical depth and 3D distributions of these properties (Capelle et al., 2014; Cuesta
et al., 2015, 2020; Vandenbussche et al., 2020; Zheng et al., 2022, 2023, 2024). Recent studies have also tested the
possibility to retrieve information on the mineralogical composition of dust based on its infrared signature (e.g., Di Biagio et
al., 2023). On the other hand, the inaccuracy of inversion algorithms in representing the infrared spectral signature of dust
aerosols can induce misinterpretations and biases in the retrieval from space of other climate-relevant parameters, such as the
temperature profile and sea surface temperature (Luo et al., 2019; Maddy et al., 2012).

In this study, we present measurements of the absorbance spectrum of mineral dust in the spectral range from 2 to 25 μm.
Results are analysed with a twofold objective: first, we experimentally investigate the significance of MIR versus FIR dust
interactions; secondly, we explore the spectral signatures of dust originating from source areas with differing mineralogical
compositions. In particular, we aim to assess the global-scale variability of dust extinction beyond the relatively well-known
MIR range and to highlight the differences or similarities between LMLD and HLD aerosols. For this purpose, thirteen dust
samples from globally distributed sources are investigated. These include eleven LMLD samples from Northern and
Southern Africa (Morocco, Niger, Chad, Namibia, Botswana), North and South America (Arizona, Chile), Middle East
(Saudi Arabia, Kuwait), and Asia (China), and two HLD samples from northern Europe (Iceland).

This paper is relevant to the forthcoming FIR satellite missions, like the Far-infrared Outgoing Radiation Understanding and
Monitoring (FORUM), which will measure for the first time the Earth's spectrum in the FIR up to 100 μm at high spectral
resolution (Palchetti et al., 2020), and their synergy with MIR observing systems, such as the Infrared Atmospheric
Sounding Interferometer (IASI) and IASI New Generation (IASI-NG) (Crevoisier et al., 2014).

## 2 Methods

This study is based on a pellet spectroscopy technique that consists in dispersing the aerosol particles in a matrix of
transparent material which is then pressed to form a homogeneous pellet. The transmission spectrum of the pellet is





measured to obtain the absorbance of the aerosol sample. To apply this technique the dust aerosols were resuspended in the laboratory from natural soil samples.

## 2.1 Parent soil selection and origin

The thirteen soil and sediment samples in this study (Table 1) were selected with the aim of representing the global scale variability of particle mineralogy. Samples, collected from different source areas worldwide, represent a depth of the first millimetres of the surface. Many of the samples had already been analysed in past laboratory studies, including simulation chamber experiments to investigate their physical, chemical and spectral optical properties at solar and MIR wavelengths (Baldo et al., 2020, 2023; Caponi et al., 2017; Di Biagio et al., 2014a, 2017, 2019, 2023).

For Northern Africa, we selected a soil from Morocco in the Northern Sahara and two samples from the Sahel, one in Niger and one in Chad (sediment from the Bodélé depression). These are important sources for medium and long-range dust transport towards the Mediterranean (Israelevich et al., 2002) and the Atlantic Ocean (Prospero et al., 2002; Reid et al., 2003). In particular, the Bodélé depression is one of the most active sources at the global scale (Goudie and Middleton, 2001; Washington et al., 2003). The two soils from the Middle East are from Saudi Arabia and Kuwait, which are important sources of dust to the Red and the Arabian seas (Prospero et al., 2002) and the North Indian Ocean (Leon and Legrand, 2003). For the second largest global source of dust, Eastern Asia, we studied one sample representative of the Gobi desert. For North and South America, soils were collected in the Sonoran Desert in Arizona and the Atacama Desert in Chile. The Sonoran Desert is a permanent source of dust in North America, the Atacama Desert is the most important source of dust in South America (Ginoux et al., 2012). For Southern Africa, we selected two soils from the Namib desert, one from the area between the Kuiseb and Ugab valleys (Namib-1, already studied in (Di Biagio et al., 2017, 2019)) and the other from the Huab ephemeral river bed (Namib-Huab), both of which are sources of dust transported towards the South-Eastern Atlantic (Vickery et al. 2013). A sample from Botswana was also selected, as an additional source from Southern Africa (Bhattachan et al., 2013, 2015). Surface sediment samples collected from two major dust hotspots in Iceland (H55, Hagavatn, and MIR45, Mýrdalssandur), which significantly contribute to the total dust emissions from this region (Arnalds et al., 2016), were also investigated. The Icelandic dust is taken as representative of HLD as Iceland is the major documented emitting high-latitude dust source so far (Meinander et al., 2022).

## 2.2 Resuspension and collection of dust aerosols from natural parent soils

Dust aerosols were re-suspended in the laboratory by mechanical shaking of the parent soil samples using the same protocol previously described by Di Biagio et al. (2017) and Baldo et al. (2023). To this aim, 5 grams of soil sample (previously sieved at 1000 µm and heated at 100° for about 1 hour) were placed in a Büchner flask and shaken at 100 Hz by means of a sieve shaker (Retsch AS200). The dust suspension in the flask was entrained by a pure $N_2$ flow at 5 L min$^{-1}$ to a glass vial



where the dust aerosol was collected. The collection time lasted for about 1 hour, whilst continuously shaking the soil, in order to collect around 1 to 2 mg of dust aerosols.

| Classification | Sample name | Collection Coordinates | Geographical zone | Country | Previous studies and references |
|---|---|---|---|---|---|
| LMLD (low and mid latitude dust) | Morocco | 31.97°N, 3.28°W | Northern Africa | Morocco | Shortwave and MIR optical properties, mineralogy (Caponi et al., 2017; Di Biagio et al., 2014a, 2017, 2019, 2023) |
| | Niger | 13.52°N, 2.63°E | Sahel | Niger | |
| | Bodélé | 17.23°N, 19.03°E | Sahel | Chad | |
| | Saudi Arabia | 27.49°N, 41.98°E | Middle East | Saudi Arabia | |
| | Kuwait | 29.42°N, 47.69°E | Middle East | Kuwait | |
| | Gobi | 39.43°N, 105.67°E | Eastern Asia | China | |
| | Arizona | 33.15 °N, 112.08°W | North America | Arizona | |
| | Atacama | 23.72°S, 70.40°W | South America | Chile | |
| | Namib-1 | 21.24°S, 14.99°E | Southern Africa | Namibia | |
| | Namib-H (Huab) | 20.92°S, 13.46°W | Southern Africa | Namibia | Formenti et al. (in prep.) |
| | Botswana | 18.36°S, 21.84°E | Southern Africa | Botswana | Baldo et al. (in prep.) |
| HLD (high latitude dust) | Iceland-M (Myrdalssandur, MIR45) | 63.54°N, 18.7°W | Northern Europe | Iceland | Shortwave optical properties, mineralogy (Baldo et al., 2020, 2023) |
| | Iceland-H (Hagavatn, H55) | 64.48°N, 20.46°W | Northern Europe | Iceland | |

**Table 1. Summary of information on the soil and sediment samples used in this study.**

**2.3 Dust pellet preparation**

Infrared transmission spectroscopy was performed by means of the pellet technique (Di Biagio et al., 2014b; Volz, 1972) using Potassium Bromide (KBr for IR spectroscopy Uvasol®, CAS No 7758-02-3, lot. B1978207 142) as transparent matrix in which dust grains collected from soil resuspension were dispersed. Pellet spectroscopy has various limitations and uncertainties to represent the optical behaviour of suspended aerosols, as discussed by Di Biagio et al. (2014b), however it

represents a reference technique to explore the infrared optical properties of aerosols and their components, without the complication of resuspension processes. A quantity of 0.5 mg of dust was weighed and diluted in 300 mg of KBr (corresponding to 0.16% of dust in KBr). Dust and KBr were weighed by means of a Mettler Toledo microbalance XPR26C whose maximum sensitivity is 1 μg. The dust-KBr mixture was mechanically shaken for few minutes to create a homogeneous mixing and slightly ground with an agate mortar. The obtained dust-KBr mixture was placed in the oven to

dry at the temperature of 110 °C for at least 2 hours and then pressed at ~9.5 Tons cm$^{-2}$ for 2 minutes to form a thin pellet. A quantity of 300.5 mg of powder (300 mg KBr plus 0.5 mg dust) was used to create a homogeneous pellet of 13 mm diameter (surface 1.33 cm$^2$) and about 1 mm thickness. All laboratory manipulations were accomplished in clean conditions in an ISO7 room. A number of 2 to 3 pellets were produced per sample depending on the amount of the collected dust aerosols (with the exception of Arizona for which only 1 pellet could be produced), to test the repeatability of the procedure, for a

total of 31 pellets. Additionally, six pure 300 mg KBr pellets were produced. All the pellets were kept in the oven at 110 °C until they were used for spectroscopy measurements in order to minimize water vapour absorption on their surface.



## 2.4 Infrared pellet spectroscopy

Absorbance spectra were recorded between 2.5 and 25 μm (4000-400 cm$^{-1}$ wavenumber) at 0.5 cm$^{-1}$ resolution by means of a PerkinElmer Spectrum Two FT-IR spectrometer. The instrument uses a Globar source, with a KBr beamsplitter and a deuterated triglycine sulphate (DTGS) detector. Pellets were placed in the spectrometer chamber. A background spectrum was acquired prior each pellet measurement and subtracted to remove the signatures of $CO_2$ and $H_2O$ possibly present in the cell. A total of 10 scans were averaged to produce the dust-KBr and the pure KBr spectra, respectively. Within the 6 spectra of pure KBr, two showed slight contamination by dust grains in the pellet and were removed from the dataset. The other 4 pure KBr spectra did not show any indication of contamination, but showed a slightly different spectral variation particularly at low wavelengths (< 5 μm), which we attribute to some degree of water vapour absorption by the highly hygroscopic KBr occurring during the pellet production, which was performed at ambient pressure instead of standard vacuum conditions. To take this effect into account, each dust-KBr spectrum was corrected by subtracting the pure KBr spectrum that best fit the baseline of each dust-KBr pellet. All the collected raw spectra for KBr and dust are shown in Fig. S1-S5 in the supplementary information. The uncertainty in the measured absorbance is less than 5% and has been estimated as the 3σ variability of the signal in the regions of no dust absorption (A< 0.01).

The pellet absorbance spectra acquired in this study are compared against in-situ absorbance spectra measured on suspended aerosols in a simulation chamber from the same LMLD source soils by (Di Biagio et al., 2017) (Fig. S6, Text S1). The comparison shows that the shape of the dust absorption in the 8-15 μm common spectral range is similar between pellets and suspended aerosols. Only the quartz band is in some cases overestimated in the pellet spectra. This is likely due to the presence of some grains of soils, much richer in quartz than the aerosols (Fig. 18 in (Adebiyi et al., 2023)), that were entrained together with the aerosol and therefore included in the pellet. This potential artefact, however, does not seem to influence the spectra in other regions within the 8-15 μm.

## 2.5 Dust mineralogical composition

The identification and quantification of the main mineral phases composing the dust aerosol particles was performed by X-Ray Diffraction (XRD) and X-ray Absorption Near-Edge Structure (XANES) analyses (Formenti et al., 2014b, a). These were performed on dust samples collected on polycarbonate filters and re-suspended applying the same generation procedure as described in Sect. 2.2. Data acquisition and analysis was presented in (Baldo et al., 2020, 2023; Caponi et al., 2017; Di Biagio et al., 2017, 2019). More detailed information on measurement procedures is provided in Text S2 in the Supplementary Information.



## 3 Results and discussion

### 3.1 Mineralogical composition of LMLD and HLD samples

The mineralogy of the thirteen analysed samples is illustrated in Fig. 1 and reported in Table S1 in the Supplementary information. The eleven LMLD samples are composed of varying proportions of clays (illite, kaolinite, chlorite, palygorskite, muscovite; 45.5-75.6% by mass), quartz (3.5-36.7%), feldspars (orthoclase, albite, microcline; 2.2-26%), calcite (2.0-21.7%) and iron oxides (hematite, goethite; 0.7-5.8 %). Iron oxides are identified in different proportions of goethite and hematite in all LMLD, except for Botswana for which no iron oxides are detected (based only on XRD analysis for this sample). Calcite is found in all samples with the exception of Niger, Bodélé and Botswana. Dolomite is detected only in the Morocco dust.

As discussed by Baldo et al. (2020), a contrasting mineralogy is obtained for the Icelandic dust samples compared to LMLD. The Iceland-M dust is mostly composed of amorphous glass material (91.3%), feldspars (anorthite; 3.5%) and pyroxene (augite; 3.6%), while Iceland-H is made of feldspars (anorthite, microcline; 53.0%), pyroxene (augite; 29.3%), olivine (forsterite; 7.2%) and minor contribution of amorphous glass (8.0%). Both Icelandic samples show the presence of iron oxides in the form of magnetite (1.4-2.0%) with lower contributions of hematite and goethite (0.2-0.5%) by XRD and Fe sequential extraction techniques. These observations are in line with analysis from same source areas in Iceland (González-Romero et al., 2024).

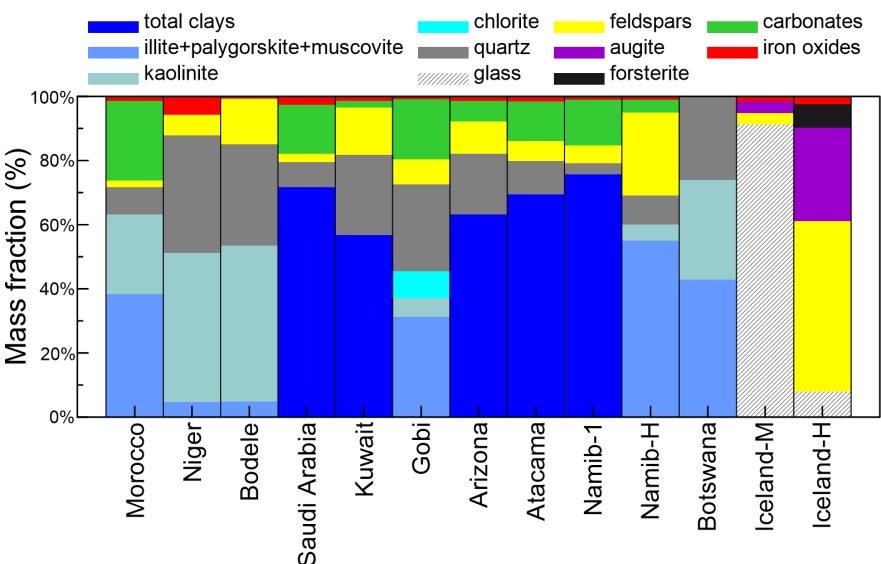

**Figure 1. Mineralogy of the thirteen aerosol samples considered in this study. The mass apportionment between the different clay species (illite, kaolinite, chlorite) for Morocco, Niger, Bodélé, and Gobi aerosols is based on literature values of the illite-to-kaolinite and chlorite-to-kaolinite mass ratios as discussed in (Di Biagio et al., 2017), while retrieved from XRD spectra analysis for Namib-H and Botswana samples. For the other samples only the total clay mass is reported. The carbonates include calcite and dolomite. Iron oxides include hematite, goethite and magnetite.**





### 3.2 Different infrared absorbance signatures in the 2.5 – 25 µm spectral range for LMLD versus HLD

The absorbance spectra measured in the 2.5-25 µm range for the thirteen LMLD and HLD samples are shown in Fig. 2. The comparison between dust absorbance and single mineral spectra is provided in Fig. 3 for Saudi Arabia, taken as an
illustrative case for LMLD due to the presence of absorption signatures from multiple minerals in its spectrum, and for Iceland-H and Iceland-M for HLD.

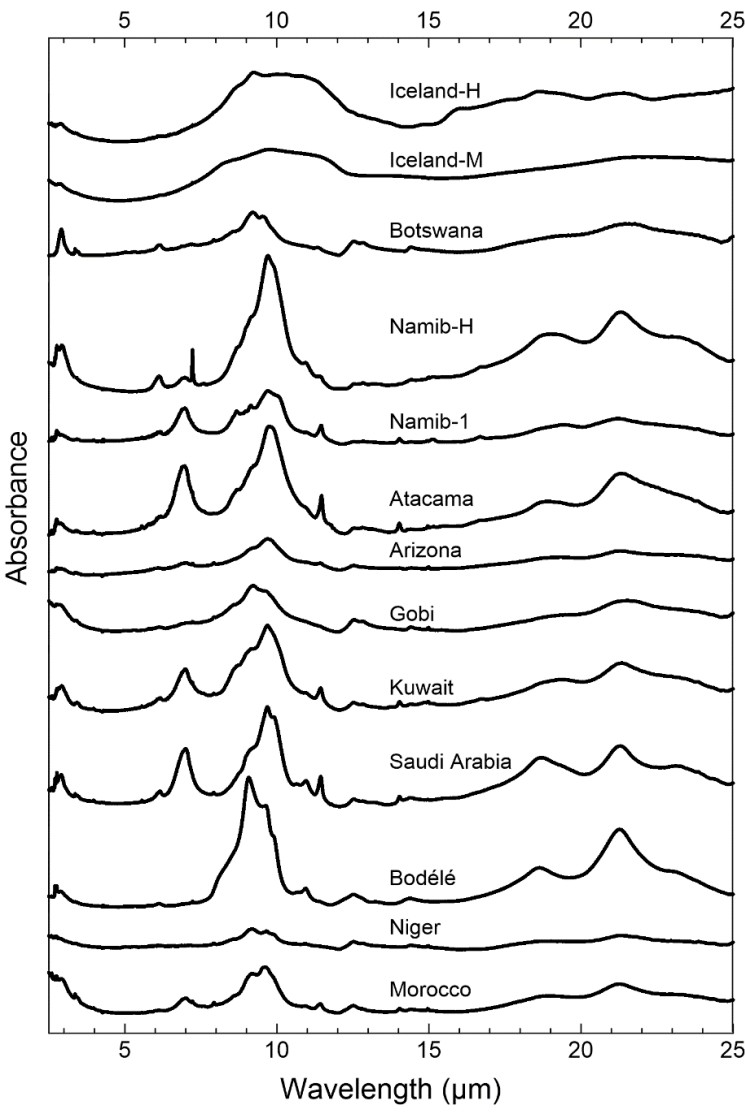

**Figure 2. Absorbance spectra measured in the spectral range 2.5–25 µm for the thirteen different dust samples in this study. Spectra are offset for improving readability. All spectra are plotted at the same scale of absorbance (0-0.4 range), with the**
**exception of Namib-H (0-0.6 range) and Bodélé (0-1.2 range). An alternative version of the figure where spectra are over-plotted is shown in Fig. S7 in the Supplementary Information.**





As depicted in Fig. 2 and 3, the absorbance spectra show large sample-to-sample variability which reflects the diversity in terms of mineral content and speciation. A marked difference is observed for the HLD compared to the LMLD samples both in respect of the position and shape of the absorption bands. The largest absorbance peaks are found for both LMLD and

HLD in the 6-12 µm and 15-25 µm regions due to the superposing contribution of multiple minerals, such as clays, quartz, carbonates and iron oxides (hematite, goethite) for LMLD and basaltic glass, feldspars, augite, forsterite and iron oxides (magnetite) for HLD. A full list of peak positions for single minerals are provided in Table S3. Note that below 5 µm the absorbance signal shows an increase for decreasing wavelength for most of the samples, which is due to the combined effect of clay absorption around 2.8 µm and residual KBr signal not completely removed through baseline correction.

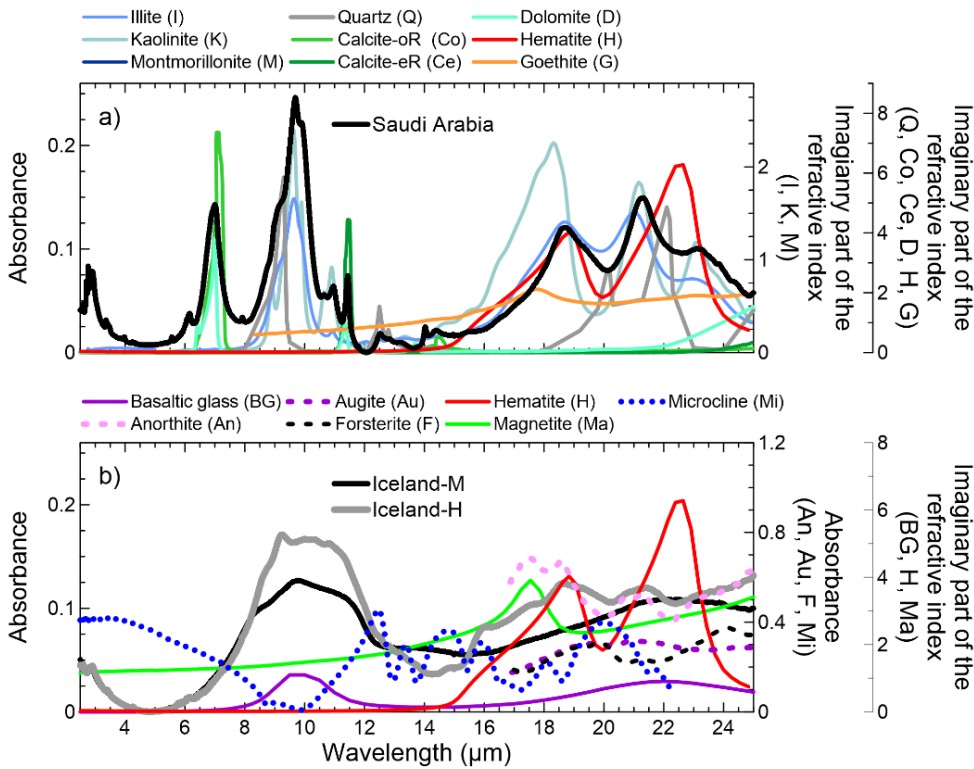


**Figure 3. Absorbance spectra measured within the spectral range 2.5–25 µm for the (a) Saudi Arabia and (b) Iceland-M and Iceland-H samples compared to the spectra of single minerals composing dust. Data for single minerals are reported as the imaginary part of the complex refractive index (continuous lines) or absorbance spectra (dashed lines). The peaks in both refractive index and absorbance spectra indicate the position of main absorption bands for the mineral. References for the plotted**
**curves and main information on single mineral data shown in this figure are provided in Table S1. The band centre wavelengths for all identified mineral absorption peaks are provided in Table S2.**

All LMLD samples display strong spectral signatures associated with clays between 9 and 11 µm, but the position and relative intensity of the peaks vary with clay speciation. The prevalence of kaolinite is reflected for instance in the presence

of the double peaks at 9.7 and 9.9 µm and the secondary peak at 10.9 µm, as identified in the Niger and Bodélé samples.





Conversely, a broader single peak located between 9.5 and 9.7 µm is identified for samples (i.e., Morocco, Arizona, Atacama, Namib-H) composed of different proportions of clay minerals (illite, kaolinite, montmorillonite, palygorskite, muscovite). The presence of quartz is associated in LMLD with a main absorption band at 9.3 µm, which induces the appearance of a more or less pronounced shoulder in the clay bands, and a secondary double peak at 12.5 and 12.8 µm, as

clearly identified in the Niger, Bodélé, Gobi, Kuwait, Arizona and Botswana samples, where quartz content is the highest (18.9-36.9% in mass). An additional region of intense absorption for LMLD is found in the 6.5-7.5 µm range associated with the specific features of carbonates (calcite, dolomite) also showing a secondary less intense peak at ~11.4 µm. The intensity of the main calcite band at 7 µm band follows the trend in calcite content (see Table S1) but it is not always proportional to it. This could suggest that differences in size-dependent mineralogy between the different samples can affect the intensity of

the absorption bands, as also discussed in (Di Biagio et al., 2014b).

Three main broad peaks are identified for all LMLD samples above 15 µm. These are contributed by superimposing bands for the different clay species between 18 and 23 µm, together with those of quartz (two bands centred at ~20 and 22 µm) and iron oxides (two large bands centred at ~19 and 23 µm for hematite, and one at ~18 µm for goethite).

For the Iceland-M sample dominantly composed of amorphous silicate (91.3%), the spectrum is mainly shaped by the two

broad absorption bands of amorphous glass centred at 9.5 and 22 µm. A third band at 14 µm is potentially associated with anorthite or augite, which are the dominant minerals by mass, but no literature spectral data for these minerals are available at these wavelengths to confirm this attribution. For the Iceland-H sample several absorption bands associated with anorthite (representing about half of the mass for this sample) are identified above 17 µm, where spectral data for this mineral are available (http://minerals.caltech.edu/FILES/Infrared_Far/Index.html) (absorption peaks centred at 18.7, 21.4, and 26.5 µm).

Microcline, augite, forsterite, and magnetite also contributed to the spectral absorbance above 17 µm and different absorption peaks of these minerals are identified, as shown in Fig. 3. The amorphous material contributes to absorption in the 8-12 µm region and above 15 µm. Small peaks are also detected in the 8-12 µm range but due to the scarcity of information on single mineral spectra, no specific attribution is possible.

The spectral diversity of HLD samples evidenced in this study is supported also by comparison with LMLD from literature

data (Di Biagio et al., 2014b; Sadrian et al., 2023), confirming the different optical signatures between low, mid and high-latitude dust. This is shown in Fig. 4 where absorbance measurements in this study are combined with literature data on pellet dust samples from worldwide locations in the 2.5-25 µm spectral range. Data include the work by Di Biagio et al. (2014b) providing measurements for five natural dust aerosol samples originated from Niger and Algeria and collected at the ground-based sites of Banizoumbou (Niger) and Tamanrasset (Algeria) in 2006 during the AMMA campaign (African

Multidisciplinary Monsoon Analysis) (Formenti et al., 2011; Rajot et al., 2008). Further spectroscopic data for 26 samples from diverse locations in USA (Arizona, California, Nevada, Utah), Africa (Chad, Botswana, Djibouti, Mali, Namibia), Spain (Las Canarias), Arabian Peninsula (Iraq, Kuwait, Qatar, Saudi Arabia), and Central and eastern Asia (Afghanistan, China) are provided by (Sadrian et al., 2023). The (Sadrian et al., 2023) dataset refers to surface soil samples sieved at 38 µm instead of aerosols. Figure 4 further illustrates the variability in the dust spectral signature across the 2.5-25 µm spectral





range for dust of additional diverse geographic origin. The LMLD samples in the present work exhibit a spectral variability

that is within the one identified in past studies and particularly in (Sadrian et al., 2023).

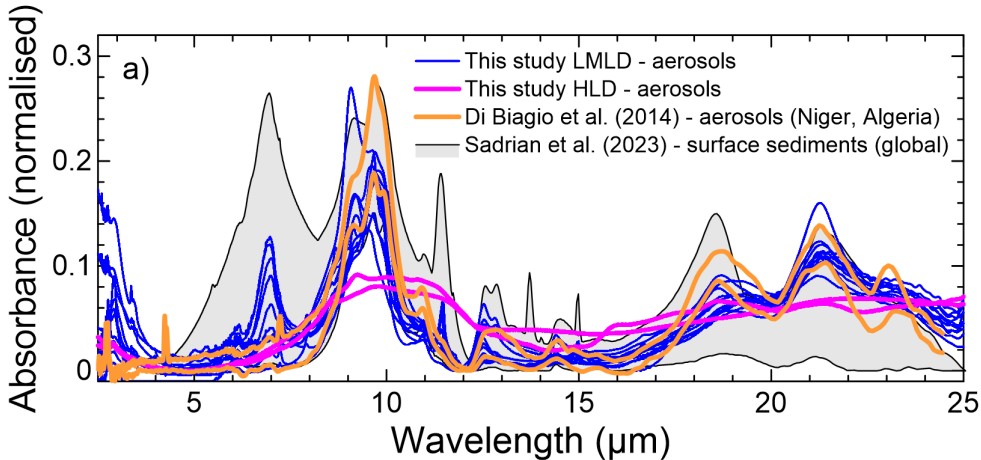

**Figure 4. Comparison of the absorbance spectra obtained in this study for LMLD (blue lines) and HLD (pink lines) and the pellet data obtained in (Di Biagio et al., 2014b) for natural aerosols in from Niger and Algeria (orange lines) and in (Sadrian et al., 2023)**
**for global surface sediments (grey shaded area). The two datasets shown for (Di Biagio et al., 2014b) represent the minimum and maximum of the absorbance measured in that study. Similarly, the shaded area envelopes the range of absorbance in (Sadrian et al., 2023). To facilitate the comparison, all data are normalized so that the integral of the absorbance is 1 in the 5-25 μm range.**

### 3.3 Relevance of MIR and FIR absorbance for LMLD and HLD

The integral of the absorbance signal in the 2.5-25 μm range for the LLMD and HLD investigated in this study and the

contributions by the MIR (2.5-15 μm) and the FIR (15-25 μm) ranges are shown in Fig. 5. The integrated spectrum in the 8-

12 μm region is also shown, as it corresponds to the MIR atmospheric window, where absorption by atmospheric gases is

relatively low, therefore of interest for atmospheric radiative transfer as most of the dust MIR signal is in this region.

The integrated absorbance in the 2.5-25 μm range is within 0.33 and 3.29 for LMLD and 1.61-1.91 for HLD. The MIR

contributes by 35-53% (LMLD) and 45-47% (HLD) to total absorbance, mostly within the 8-12 μm region, while the FIR

contribution is within 47-65% (LMLD) and 53-55% (HLD), representing more than half of the total integrated absorbance.

Literature data on natural dust samples in Fig. 4 also confirm the relevance of the FIR absorbance for dust of varying origin.

The integrated absorbance in the 2.5-25 μm is within 0.44 and 1.0 for the samples in (Di Biagio et al., 2014b), and within

0.41 and 5.50 (with an average value of 1.7 ± 1.2) in the 4-25 μm for the 26 samples in (Sadrian et al., 2023).The FIR signal

above 15 μm contributes 50-60% of the integrated absorbance in (Di Biagio et al., 2014b), and 28-63% in (Sadrian et al.,

2023). Only one sample in (Sadrian et al., 2023) displays a low contribution of the FIR (5%) to the absorbance signal.





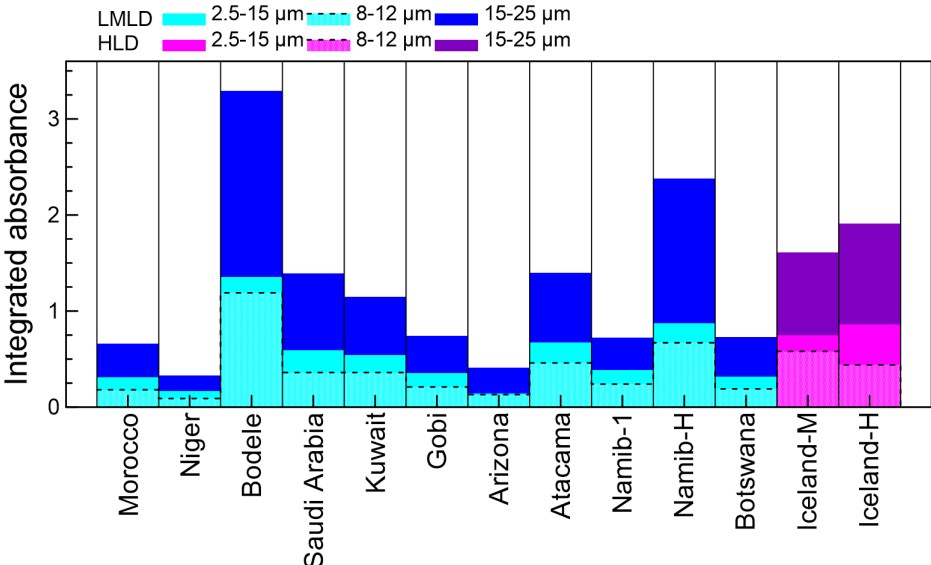

**Figure 5. Integral of the absorbance signal in the 2.5-25 µm range for the LLMD and HLD investigated in this study. The contributions by the MIR (2.5-15 µm), the FIR (15-25 µm) and the 8-12 µm region are shown.**


The few existing pellet spectroscopic measurements of natural dust aerosols beyond 25 µm (Fouquart et al., 1987; Volz, 1972, 1973; reported as imaginary part of the refractive index in Supplementary Fig. S8), despite not allowing to explore the diversity of dust absorption from global sources, support anyhow the presence of a significant absorption signal continuously up to 40 µm. These data were acquired at low spectral resolution and correspond to rainout dust samples collected in

Germany (Volz, 1972), Saharan dust from Barbados (Volz, 1973) and Niger dust (Fouquart et al., 1987).

The relevance of dust absorption in the FIR for sources of varying mineralogy is further evidenced when looking at the spectra of single minerals composing LMLD and HLD. Literature data of the imaginary part of the refractive index or absorbance spectra for single dust minerals extending through the FIR range up to 100 µm are shown in Fig. 6. Information on the datasets and the band centre wavelengths for all identified mineral absorption peaks are provided in Tables S2 and S3

in the Supplementary Information. As shown in Fig. 6, all minerals show multiple and often superposing absorption bands in the 25-100 µm wavelength range, therefore supporting the relevance of dust-radiation interaction in the FIR well beyond 25 µm and 40 µm, as measured for natural dust samples. The strongest peaks above 25 µm are identified for clays (kaolinite, illite; 28.8, 36.2, 51.3, 91.7 µm), iron oxides (hematite, goethite, magnetite; 22.6, 28.6 29.1, 33.5, 37.3 µm), and carbonates (calcite, dolomite; 28.5, 33.0, 38.5, 45.9, 64.0). Feldspars (albite, orthoclase, anorthite), pyroxene (augite), and olivine

(forsterite) show several absorption peaks in the broad range 26 to 77 µm. The intensity of the absorption peaks is of comparable intensity, and in some cases higher, than those below 25 µm, particularly for hematite, calcite and dolomite.

On the other hand, for several minerals such as basaltic glass, quartz and microcline, information is lacking in the FIR. Figure 6, while showing the potential relevance of dust absorption throughout the FIR domain, also highlights the paucity of spectral data available for single dust mineral components, which sum up to the low available information for natural dust.



For most of minerals, the data do not cover the full infrared spectral range, being limited either to the MIR or to part of the FIR. Information on the complex refractive index are available only for some minerals, while no data exist for feldspars, pyroxene and olivine. Almost all data available have been acquired for compressed pellets or often crystals, so they provide limited representativeness for suspended dust.

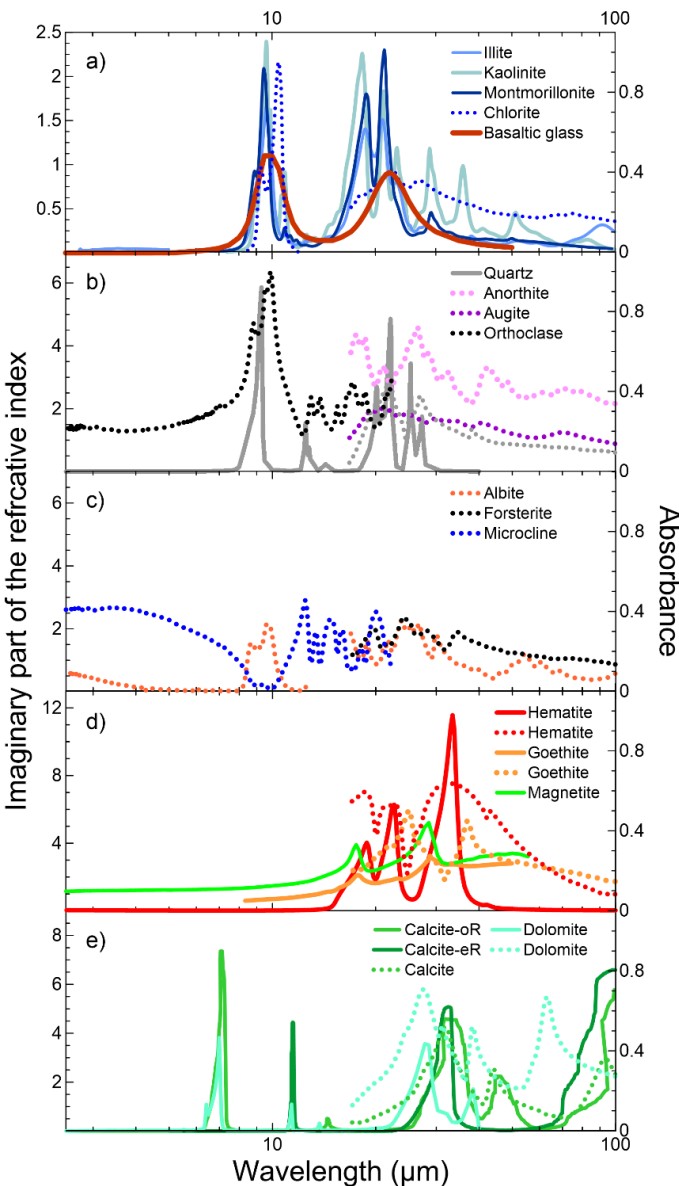

**Figure 6. Imaginary refractive index and absorbance spectra within the spectral range 5–100 µm for individual minerals composing low and high latitude dust. References for the plotted curves are provided in Table S1. Data are either reported as imaginary refractive index (continuous lines) or absorbance spectra (dashed lines). Original data for orthoclase (panel b) and albite (panel c) have been scaled to ease comparison.**



**3.4 Implications for dust direct radiative effect and infrared remote sensing**

The results from this study show that the absorbance of dust in the FIR up to 25 µm is comparable in intensity to that in the MIR. These observations highlight the potential important contribution of FIR interactions to the total dust direct radiative effect for low to high latitude dust sources. As a matter of fact, the FIR range is at present not specifically considered in model simulations due to the lack of knowledge on the dust optical properties at these wavelengths. However, it can be climatically relevant in different regions, notably dry areas, as well as high latitudes where infrared radiation control the

radiative budget over large part of the year.

Another observation from the present analysis is the comparable intensity of the infrared absorbance, both in the MIR, FIR and the 8-12 µm window, for the HLD compared to the LMLD samples. The integrated absorbance of HLD is indeed at the upper end of that measured for low and mid-latitude dust. In a previous study, (Baldo et al., 2023) estimated that Icelandic dust absorption in the solar range (370-590 nm) is at the upper end of typical values for LMLD as reported by (Di Biagio et

al., 2019), while in the near infrared (660-950 nm) the absorption by Icelandic dust is up to 2-8 times higher than most of dust aerosols from northern Africa and Asia. The results from (Baldo et al., 2023) combined with observation from this work suggest the relevance of dust-radiation interaction for HLD across the whole atmospheric spectrum.

Data from this study also highlight the variability of dust spectral absorbance as measured up to 25 µm for dust of diverse origin and composition, particularly when contrasting LMLD versus HLD types. Specific minerals are identified to modulate

the intensity and the spectral variation of dust absorption bands, producing characteristic spectral signatures for LMLD (clays, carbonates, quartz, iron oxides) and HLD (amorphous silicate, anorthite, microcline, augite, forsterite). This suggests that the spectral signature of dust in the MIR and FIR could be used to identify mineralogical composition for different dusts, which can be used to differentiate the origin of airborne particles based on remote sensing infrared observations, such as those from IASI/IASI-NG and FORUM. Single mineral spectra also suggest the presence of additional specific mineral

signatures beyond 25 µm, which supports further possibilities to detect and characterise global dust from ground-based and space-borne hyperspectral remote sensing observations in the FIR. Indeed, as demonstrated in (Di Biagio et al., 2023), infrared spectral signatures can be used to derive quantitative information of dust mineralogy, which enables to distinguish dust sources and potentially follow dust plume evolution transport in the atmosphere.

**4 Conclusions and perspectives**

In this work we explored the dust spectral signature and sample-to-sample variability of absorbance in the infrared region including the MIR and the FIR up to 25 µm. To this aim we investigated thirteen aerosol samples generated in the laboratory from surface soils and sediments originated from global dust source regions including both low, mid and high-latitude dust source areas from four continents (Africa, Asia, America, Europe).





The analysis of absorbance spectra acquired in the present study, corroborated with past literature results on natural dust and
single minerals, evidences the significance of dust-radiation interactions in the infrared range for both LMLD and HLD and
the relevant contribution of the FIR to total dust infrared absorption. The FIR is estimated to represent up to 65% of total
infrared absorbance as measured up to 25 µm in the present work. However its consideration in radiative transfer models is
hampered by lack of data on dust optical properties in this spectral range. Extending investigation of the dust spectral optical
properties (absorbance spectra, mass absorption and extinction cross section, complex refractive index) to the FIR is
necessary to improve the understanding of the role of dust in the radiative transfer and in the regional and planetary radiative
budget. New knowledge should include both natural dust samples and single minerals composing dust, as for both the
knowledge is scarce and as both are relevant to improve representation of dust in regional and global models (Gómez
Maqueo Anaya et al., 2024; Gonçalves Ageitos et al., 2023; Li et al., 2024; Obiso et al., 2024; Scanza et al., 2015). Future
measurements should preferentially focus on suspended natural samples instead of pellets, or even crystal samples as for
single minerals, to represent closely the natural state of mineral dust aerosols.

Our results also put in evidence the diversity in infrared spectral signature for HLD samples compared to LMLD due to
differences in mineralogical composition. These differences are of particular significance for remote sensing applications as
they suggest the possibility to distinguish the composition and the origin of dust based on infrared hyperspectral
observations. It is worth noting that the present analysis is limited to two Icelandic dust sources only. We have taken Iceland
as representative of HLD as one of the strongest sources of dust at high latitudes identified so far (Arnalds et al., 2016;
Dagsson-Waldhauserova et al., 2017; Meinander et al., 2022). Previous studies have also focussed on Icelandic dust
highlighting differences in its composition and optical behaviour at solar wavelengths compared to African and Asian dust
(Baldo et al., 2020, 2023; González-Romero et al., 2024). The results in the present work, combined with these past studies,
hence underscore the importance of extending investigation of HLD composition and spectral optical properties from other
relevant sources of HLD in the Northern and Southern Hemisphere, including Greenland, Canada, Svalbard, Patagonia, and
Antarctica (Meinander et al., 2022, 2025). Although HLD represents only about 1% of global dust emissions to date
(Meinander et al., 2022), the role of HLD in the polar environment is expected to progressively increase in the next years and
decades, because of emissions increase in a warming climate due to both increasing exposure of natural sources (reduction of
glaciers and ice– and snow–covered surfaces for a growing fraction of the year) and rising impact of anthropogenic activities
(mines, road dust) (Thaarup et al., 2020). Ice core data show that there is a significant increase in Greenland dust loading
since 2000, highly correlated with the local air temperature (Amino et al., 2021). As emphasized in (Kok et al., 2023), model
schemes should start explicitly account for dust emissions from high latitudes, and our results underline the need for source-
specific characterization.




**Data availability**

The retrieved absorbance spectra from this study are available at the EasyData data portal https://www.easydata.earth/#/public/home with the following doi number: https://doi.org/10.57932/905eff0b-d508-4aad-

a422-5708e3132790. The single mineral spectra used in this study (Fig. 3, Fig. 6) are available through original publications or open access repositories as listed in Table S2. The data from (Sadrian et al., 2023) are available as Supplementary Information from their paper.

**Author contributions**

CDB, PF and PS designed the experiments and discussed the results. EB conducted the experiments with contributions by CDB, SC, AB, MC, and EP. EB and AO performed the data analysis of the pellet spectroscopy measurements under the supervision of CDB. EB, CDB, and PF collected single mineral data. CB, SL, and SN contributed to the mineralogical analyses. MA, PDW, KD, KK, JK, AC, GO, SP, TS, DS, and ZS collected and shared the soil samples used for the experiments. PS, CDB, and PF provided funding acquisition and administration of the project. CDB wrote the manuscript.

All authors reviewed and commented on the paper.

**Competing interests**

The authors declare that they have no competing interests.

**Acknowledgements**

Contributions to experimental work by Gael Noyalet, Manuela Cirtog, Marc David, Juan Cuesta, Stephane Alfaro, Bernadette Chatenet, Beatrice Marticorena, and Jean Louis Rajot at Laboratoire Interuniversitaire des Systemes Atmospehriques (LISA, UMR7583 CNRS) are gratefully acknowledged.

**Financial support**

This work has been supported by the Centre National d'Études Spatiales (CNES) via TOSCA/FORUM grant. K. Kandler was funded by the Deutsche Forschungsgemeinschaft (DFG, German Research Foundation) – 416816480; 417012665. A. Chaput was supported by Mitacs Globalink Research.

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
