# Peer review of "Mid- and Far-Infrared Spectral Signatures of Mineral Dust from Low- to High-Latitude Regions: significance and implications"

_EGUsphere, 2025_

## Author Comment (AC1)

**Revisions of the paper**

**Mid- and Far-Infrared Spectral Signatures of Mineral Dust from Low- to High-Latitude Regions: significance and implications**

Claudia Di Biagio1, Elisa Bru2, Avila Orta2, Servanne Chevaillier2, Clarissa Baldo1,2, Antonin Bergé1,a, Mathieu Cazaunau2, Sandra Lafon1, Sophie Nowak3, Edouard Pangui2, Meinrat O. Andreae4,5,6, Pavla Dagsson-Waldhauserova7,8, Kebonyethata Dintwe9, Konrad Kandler10, James S. King11, Amelie Chaput11,b, Gregory S. Okin12, Stuart Piketh13, Thuraya Saeed14, David Seibert15, Zongbo Shi16, Earle Williams17, Pasquale Sellitto2,18, Paola Formenti1

First of all we would like to thank the reviewers for their careful reading of the paper and their helpful comments. The following provides a point-by-point response to the reviewer's comments (comments in black, answers in blue).

To note that in the revised version of the manuscript we have revised the colour scheme and line style for all the figures, included those in the supplementary, in order to ensure readers with colour vision deficiencies to correctly interpret our findings.

We have also adjusted some minor points in the text to ease reading and add some precisions.

**Point-by-point response to reviewers**

**Reviewer 1**

This paper articulates and fills a clear gap in the literature: the scarcity of measurements of mineral dust optical properties beyond a wavelength of ~15 um. The paper is also well written and the methodology rigorous, following previous very well-regarded work by the main authors. I do have two important comments about the framing of the paper that should be addressed before publication.

**Major comments:**

• The authors take two samples of Icelandic dust to be representative of high-latitude dust. However, because Iceland is a volcanic island the mineralogical composition of its dust is likely to be quite different (more basaltic and darker) than that of other high-latitude sources, which are normally more silicate and quartz rich. The mineralogy of Icelandic dust might thus be an outlier among high-latitude dust sources and should not be taken as representative of high-latitude dust sources. I recommend that the authors replace "high latitude dust" with "Icelandic dust" in the title, abstract, and so forth. If the authors do want to take the Icelandic dust as representative then they should provide evidence that this is reasonable to do, for instance from a comparison of the mineralogy of Icelandic dust to that of other high-latitude sources.

We thank the reviewer for this comment. As explained in Section 2.1, Icelandic dust was chosen due to the fact that Iceland is the major documented emitting high-latitude dust source so far. Given the large areas covered in the high latitudes and the wide-ranging geological origins, we agree that it may not be realistic to find samples that are "representative" of high latitude dust. Therefore, we modified the text at specific points where it was mentioned that Iceland is "representative of HLD" (Section 2.1, line 124; Conclusions, line 365). Further changes applied to the text are:

Abstract, line 42 : we replaced HLD with "Icelandic dust"

- We changed the title of Sections 3.1, 3.2 and 3.3 by replacing HLD with "Icelandic dust"
- We changed at several points throughout the text the word "HLD" with "Icelandic dust" or "Icelandic HLD".

Our title, on the other hand, does not imply that Icelandic dust are representative of high latitude dust, but it acknowledges the fact that investigated samples are from both low and high latitudes. So we kept it unchanged.

Worth to note however that other examples of very active volcanic dust sources are documented at high latitudes, including Alaska, Canada (Kluane Lake), Antarctica (Antarctic Peninsula, McMurdo valleys, parts of Patagonia's dust sources), suggesting that Icelandic dust is possibly not an outlier among HLD sources (Bullard et al., 2016; Meinander et al., 2022). However, very little knowledge still exists on the chemical and mineralogical properties of HLD, with the exception of a few source areas and more investigation is required to understand the regional-scale variation of the physico-chemical properties of HLD. We mention this aspect in the Conclusions.

The authors state at various locations in the manuscript that the optical properties in the far infrared region (beyond 15 um) are important to climate and remote sensing. However, conventional wisdom is that (dust) absorption (well) outside of the atmospheric window does not matter much because the atmosphere is very opaque beyond 15 um due to abundant absorption by water vapor. So I have a hard time imagining that the dust interaction with radiation would normally matter much for Earth's radiation budget. Two exceptions worth highlighting are for dust in the upper troposphere and for dust in the Arctic (mentioned in the paper), but the dust concentration is very low in both locations. A clearer argument could be made for remote sensing, as long as it's made clear that this would only work in the upper troposphere / stratosphere and the dry poles. If the authors do want to make the point that dust interactions with radiation beyond 15 um matter for Earth's climate, I recommend they show (1) Earth's outgoing radiation as a function of wavelength and (2) the height in the atmosphere at which the optical depth to TOA equals 1 (the emission height) for one or more standard profiles (e.g., mid-latitude summer). That can make at least a qualitative argument that the dust optical properties for wavelengths larger than 15 um matter for Earth's radiation budget.

Concerning the dust relevance for the radiative budget and remote sensing, we fully agree on the necessity to investigate the dust interactions beyond 15 µm taking carefully into account the Earth's spectrum and different atmospheric conditions. A paper fully dedicated to investigate this topic is in preparation and expected to be submitted in the next few weeks to Atmos. Meas. Tech. (Sellitto, P. et al. "The sensitivity of the Far-infrared Outgoing Radiation Understanding and Monitoring (FORUM) mission to dust aerosols: a pseudo-observations analysis"). In this new work we will focus particularly on the sensitivity of remote sensing observations to dust aerosols, their varying mineralogical composition, spectral optical properties, atmospheric load and vertical distribution. In the revised version of the paper (Section 3.4) we therefore refer to this forthcoming submission to direct the reader to the specific analysis of this important aspect (new text in bold):

"Data from this study also highlight the variability of dust spectral absorbance as measured up to 25 µm for dust of diverse origin and composition, particularly when contrasting LMLD versus Icelandic HLD types. Specific minerals are identified to modulate the intensity and the spectral variation of dust absorption bands, producing characteristic spectral signatures for LMLD (clays, carbonates, quartz, iron oxides) and Icelandic dust (amorphous silicate,

anorthite, microcline, augite, forsterite). This suggests that the spectral signature of dust in the MIR and FIR could be used to identify mineralogical composition for different dusts, which can be used to differentiate the origin of airborne particles based on remote sensing infrared observations, such as those from IASI/IASI-NG and FORUM. Single mineral spectra also suggest the presence of additional specific mineral signatures beyond 25 µm, which supports further possibilities to detect and characterise global dust from ground-based and space-borne hyperspectral remote sensing observations in the FIR. Indeed, as demonstrated in Di Biagio et al. (2023), infrared spectral signatures can be used to derive quantitative information of dust mineralogy, which enables to distinguish dust sources and potentially follow dust plume evolution transport in the atmosphere. A specific analysis of the sensitivity of MIR and FIR remote sensing to dust aerosols is provided by Sellitto et al. (in preparation). »

Concerning the relevance of FIR interactions for the radiative budget, as noted by the reviewer we already specify the potential relevance specifically for dry areas and the Arctic. However we slightly modified the first paragraph in Section 3.4 also to explicitly mention dust in the upper troposphere as well (new text in bold):

"The results from this study show that the absorbance of dust in the FIR up to 25 µm is comparable in intensity to that in the MIR. These observations highlight the **potential** contribution of FIR interactions to the total dust direct radiative effect for low to high latitude dust sources. As a matter of fact, the FIR range is at present not specifically considered in model simulations due to the lack of knowledge on the dust optical properties at these wavelengths. However, it can be climatically relevant in different regions, notably dry areas and the upper troposphere, as well as high latitudes where infrared radiation control the radiative budget over large part of the year."

**Minor comments:**

• Line 68: the 88% here seems very specific. My recollection of these papers also is that the error bar on the net DRE overlapped with zero, meaning that the dust-radiation interactions in the infrared could be >100% of those in the shortwave spectrum.

The sentence has been modified as follows (new text in bold):

"Recent modelling efforts including state-of-the-art representation of dust size distribution and spectral scattering and absorption from the ultraviolet to the mid-infrared MIR (3-15  $\mu$ m) suggest that at the global and annual scale the dust-radiation interactions in the infrared can counteract a significant fraction and even fully offset the negative DRE at TOA (Di Biagio et al., 2020; Kok et al., 2017; Wang et al., 2024)."

• Many citations that are part of the sentence are listed with parentheses, which is a bit tedious to read. For instance, line 263 should probably be "The Sadrian et al. (2023) dataset..." instead of "The (Sadrian et al., 2023) dataset..."

The citation style has been amended in line with the reviewer's suggestion.

**Reviewer 2**

The paper discusses the importance of further measurements of dust spectral properties in the mid-infrared ( $\sim$ 3–15 µm) and, particularly, in the far-infrared ( $\sim$ 15–100 µm) regions, as they may have a substantial effect on total dust direct radiative forcing and climate model estimations. The study uses 13 dust samples from low-, mid-, and high-latitude regions, employing dust-KBr pellets to obtain absorbance spectra. The results show that the spectral signatures of low- and mid-latitude dust are in good agreement with past studies. However, dust collected from high-latitude regions exhibits different spectral characteristics due to its distinct composition, which is primarily dominated by amorphous minerals.

This study makes a valuable contribution by adding new dust spectral data from low-, mid-, and high-latitude regions to the global library of mineral dust spectra. As more remote sensing instruments are being developed to cover the mid- and far-infrared spectra and are planned to monitor aerosol particles, studies like this can significantly help reduce uncertainties in estimating the radiative effects of dust and in quantifying their abundance through remote sensing retrievals.

**Comments:**

• For Figure 2: Since the absorbance spectra for the samples have already been offset, you can label the y-axis as "Absorbance (offset for clarity)." This is a more traditional approach and improves the overall clarity of the plot.

**Figure 2 has been changed accordingly.**

For line 245: where it is mentioned "A third band at 14  $\mu$ m is potentially associated with anorthite or augite, which are the dominant minerals by mass, but no literature spectral data for these minerals are available at these wavelengths to confirm this attribution." ,You can find the transmission spectra of pure minerals (~ 2.5-25 um), including anorthite and augite, for comparison purposes in: Salisbury, J. W., Walter, L. S., Vergo, N., & Daria, D. M. (1991). Infrared (2.1–25  $\mu$ m) spectra of minerals. Johns Hopkins University Press. ISBN: 978-0801844232.

We thank the reviewer for pointing at this reference. Transmission data for anorthite and augite are provided in form of plots in Salisbury et al. (1987) and Salisbury et al. (1991). Analysis of transmission spectra support the presence of an absorption band at ~14  $\mu$ m that - by comparison with Salisbury et al. data - seems mostly associated to anorthite. We have then changed the text accordingly (new text in bold):

"For the Iceland-M sample dominantly composed of amorphous silicate (91.3%), the spectrum is mainly shaped by the two broad absorption bands of amorphous glass centred at 9.5 and 22  $\mu$ m. A third band at 14  $\mu$ m is identified and potentially associated with anorthite (Salisbury et al., 1987, 1991; data not shown). For the Iceland-H sample several absorption bands of anorthite (representing about half of the mass for this sample) are identified, with the absorption peaks centred at 18.7, 21.4, and 26.5  $\mu$ m. Microcline, augite, forsterite, and magnetite also contributed to the spectral absorbance above 17  $\mu$ m and different absorption peaks of these minerals are identified, as shown in Fig. 3. The amorphous material contributes to absorption in the 8-12  $\mu$ m region and above 15  $\mu$ m. Small peaks are also detected in the 8-12  $\mu$ m range but due to the scarcity of information on single mineral spectra, no specific attribution is possible."

• That would be good if you could remove the white columns on both sides of Fig. 1.

Figure 1 has been changed accordingly. For consistency, the Figure 5 has been also modified following the reviewer's suggestion.

• Line 275: edit LLMD. It should be LMLD.

Corrected.

**References**

Bullard, J. E., Baddock, M., Bradwell, T., Crusius, J., Darlington, E., Gaiero, D., Gassó, S., Gisladottir, G., Hodgkins, R., McCulloch, R., McKenna-Neuman, C., Mockford, T., Stewart, H., and Thorsteinsson, T.: Highlatitude dust in the Earth system, Reviews of Geophysics, 54, 447–485, https://doi.org/10.1002/2016RG000518, 2016.

Di Biagio, C., Balkanski, Y., Albani, S., Boucher, O., and Formenti, P.: Direct Radiative Effect by Mineral Dust Aerosols Constrained by New Microphysical and Spectral Optical Data, Geophysical Research Letters, 47, e2019GL086186, 2020.

Di Biagio, C., Doussin, J.-F., Cazaunau, M., Pangui, E., Cuesta, J., Sellitto, P., Ródenas, M., and Formenti, P.: Infrared optical signature reveals the source–dependency and along–transport evolution of dust mineralogy as shown by laboratory study, Scientific Reports, 13, 13252, https://doi.org/10.1038/s41598-023-39336-7, 2023.

Kok, J. F., Ridley, D. A., Zhou, Q., Miller, R. L., Zhao, C., Heald, C. L., Ward, D. S., Albani, S., and Haustein, K.: Smaller desert dust cooling effect estimated from analysis of dust size and abundance, Nature Geoscience, 10, 274–278, https://doi.org/10.1038/ngeo2912, 2017.

Meinander, O., Dagsson-Waldhauserova, P., Amosov, P., Aseyeva, E., Atkins, C., Baklanov, A., Baldo, C., Barr, S. L., Barzycka, B., Benning, L. G., Cvetkovic, B., Enchilik, P., Frolov, D., Gassó, S., Kandler, K., Kasimov, N., Kavan, J., King, J., Koroleva, T., Krupskaya, V., Kulmala, M., Kusiak, M., Lappalainen, H. K., Laska, M., Lasne, J., Lewandowski, M., Luks, B., McQuaid, J. B., Moroni, B., Murray, B., Möhler, O., Nawrot, A., Nickovic, S., O'Neill, N. T., Pejanovic, G., Popovicheva, O., Ranjbar, K., Romanias, M., Samonova, O., Sanchez-Marroquin, A., Schepanski, K., Semenkov, I., Sharapova, A., Shevnina, E., Shi, Z., Sofiev, M., Thevenet, F., Thorsteinsson, T., Timofeev, M., Umo, N. S., Uppstu, A., Urupina, D., Varga, G., Werner, T., Arnalds, O., and Vukovic Vimic, A.: Newly identified climatically and environmentally significant high-latitude dust sources, Atmos. Chem. Phys., 22, 11889–11930, https://doi.org/10.5194/acp-22-11889-2022, 2022.

Salisbury, J. W., Walter, L. S., and Vergo, N.: Mid-infrared (2.1-25 um) spectra of minerals; first edition, Open-File Report, U.S. Geological Survey, https://doi.org/10.3133/ofr87263, 1987.

Salisbury, J. W., Walter, L. S., Vergo, N., and Diana, D. M.: Infrared (2.1–25  $\mu m$ ) spectra of minerals, 1991.

Wang, H., Liu, X., Wu, C., Lin, G., Dai, T., Goto, D., Bao, Q., Takemura, T., and Shi, G.: Larger Dust Cooling Effect Estimated From Regionally Dependent Refractive Indices, Geophysical Research Letters, 51, e2023GL107647, https://doi.org/10.1029/2023GL107647, 2024.